# Treatment Patterns and Outcomes of Patients with Grade 4 Glioma Treated with Radiation during the COVID-19 Pandemic

Manik Chahal [1,*], Ghufran Aljawi [2], Rebecca Harrison [1], Alan Nichol [2] and Brian Thiessen [1]

1   Division of Medical Oncology, British Columbia Cancer-Vancouver Cancer Centre, Vancouver, BC V5Z 4E6, Canada
2   Division of Radiation Oncology, British Columbia Cancer-Vancouver Cancer Centre, Vancouver, BC V5Z 4E6, Canada
*   Correspondence: manik.chahal@phsa.ca

**Abstract:** During the first year of the COVID-19 pandemic there was a global disruption in the provision of healthcare. Grade 4 gliomas are rapidly progressive tumors, and these patients are at risk of poorer outcomes due to delays in diagnosis or treatment. We retrospectively evaluated the impact of the pandemic on treatment patterns and outcomes of patients with grade 4 gliomas in British Columbia. We identified a cohort of 85 patients treated with radiotherapy between March 2020–2021 (COVID era) and compared baseline characteristics, treatments, and outcomes with a control cohort of 79 patients treated between March 2018–2019 (pre-COVID era). There were fewer patients treated with radiotherapy over age 65 in the COVID era compared to the pre-COVID era ($p = 0.037$). Significantly more patients were managed with biopsy relative to partial or gross total resection during the COVID era compared to the pre-COVID era ($p = 0.04$), but there were no other significant differences in time to assessment, time to treatment, or administration of adjuvant therapy. There was no difference in overall survival between eras ($p = 0.189$). In this assessment of outcomes of grade 4 gliomas during the pandemic, we found that despite less aggressive surgical intervention during the COVID era, outcomes were similar between eras.

**Keywords:** high-grade glioma; grade 4 glioma; glioblastoma; COVID-19; management; outcomes

## 1. Introduction

During the first year of the COVID-19 pandemic there was a notable global disruption in the provision of healthcare. With significant pressure on hospital resources, the pandemic had a secondary indirect effect on non-COVID patients. Given the resource-intensive nature of cancer care, patients with cancer were particularly vulnerable to disruption in management [1]. Within Canada, surgical slowdowns were instituted to help conserve healthcare resources. A microsimulation model based on real-world population data on cancer care that estimated wait times for cancer surgeries suggested that pandemic-related slowdowns were projected to result in decreased long-term survival for many patients with cancer [2]. Furthermore, with an increased risk in cancer patients of developing complications from COVID-19 due to their immunosuppressive status [3–5], there were additional concerns regarding balancing competing risks between COVID-19 and the primary malignancy.

Grade 4 glioma including glioblastoma is the most common and most aggressive primary intracranial malignancy. Patients with glioma are especially vulnerable to potential side effects from COVID-19 due to a tendency towards older age, increased risk of functional impairment due to neurocognitive decline, and increased use of corticosteroids thereby causing further immunosuppression and greater susceptibility to infections [6,7]. COVID-19 infection during the early stages of the pandemic was also known to have neurological sequelae, with CNS infection hypothesized to be mediated via angiotensin-converting enzyme receptors present on endothelial, glial, and neuronal cells [8]. It was

therefore also suspected that due to having a pre-existing injury to the brain, patients with glioma were more susceptible to the neurologic side effects of COVID infection [9].

Given the high morbidity and mortality of grade 4 glioma, a number of management recommendations were published to help guide physicians during the first year of the pandemic when best practices are unavailable, impractical, or unsafe [7,10,11]. Recommendations have been continually updated and different centers have adapted their own approaches based on their own experiences with COVID-19. The optimal first-line treatment involves maximal surgical resection, and concurrent chemoradiotherapy for 6 weeks, followed by adjuvant monthly chemotherapy with temozolomide (TMZ) [12]. Some guidelines recommend hypofractionated radiotherapy regimens for patients at high risk of severe illness to reduce the exposure of patients to COVID-19 by reducing time at healthcare facilities [10]. Other guidelines have also suggested that the addition of TMZ chemotherapy must be individualized on a case-by-case basis and balanced against the risk of complications [11].

During the COVID-19 pandemic, efforts were made to delay non-urgent cancer surgeries and minimize hospital visits to mitigate the risk of COVID infection in the vulnerable cancer population. In patients with primary high-grade brain tumors who are at risk of rapid progression and for whom daily visits for chemoradiation therapy are standard of care, these efforts could have a marked impact on care delivery. We, therefore, performed this retrospective cohort analysis of patients across the six provincially regulated tertiary cancer centers in British Columbia to evaluate the outcomes of patients with high-grade glioma during the COVID-19 pandemic and to identify delays or alterations in management that may have influenced these clinical outcomes.

## 2. Materials and Methods

### 2.1. Study Population

We conducted a retrospective cohort study of patients aged 18 or older with histologically proven grade 4 glioma (IDH-mutated grade 4 astrocytoma and IDH-wildtype glioblastoma) who were treated with external beam radiotherapy (EBRT) between 1 March 2020–1 March 2021 (COVID era), and a control cohort treated between 1 March 2018–1 March 2019 (pre-COVID era) at the 6 provincially regulated tertiary care British Columbia (BC) Cancer centers. Diagnosis was established by surgical resection and pathology (based on histologic and molecular analyses). Patients were excluded if they had a diagnosis other than grade 4 glioma (including no pathologic diagnosis), declined treatment, or were lost to follow-up. All patients included had been prescribed radiotherapy. The study was approved by institutional research ethics board.

### 2.2. Treatment

All patients were evaluated by neurosurgeons and underwent tumor biopsy with or without maximal safe resection to confirm cancer diagnoses. Extent of surgery was determined by surgical operating notes indicating whether all visible tumor was resected (gross total resection), if there was visible tumor tissue remaining (subtotal resection), or if biopsy alone was performed.

EBRT technique was photon beam delivered with intensity-modulated radiotherapy. Image-guided RT (IGRT) for all patients was performed using cone beam CT. Choice of radiotherapy dose was dependent on patient's performance status and age. The prescribed dose ranged from 2500–6000 cGy. Clinical target volume (CTV) is defined by 2 cm expansion around the enhancing gross tumor volume and surgical bed on T1 magnetic resonance imaging sequence. Planning target volume (PTV) defined by 3 mm expansion around the CTV; PTV is to be covered by $\geq$95% of the prescription dose.

When prescribed, standard concurrent temozolomide was given for six weeks during EBRT at 75 mg/m$^2$/day followed by adjuvant temozolomide for six maintenance cycles at a dose of 150–200 mg/m$^2$/day for 5 days of a 28-day cycle [12].

## 2.3. Data Collection

Chart review of electronic medical records was performed to collect patients' baseline characteristics including age, sex, comorbidities, and ECOG performance status. Tumor specifics of IDH mutational status, MGMT methylation status, and average mass diameter (calculated by averaging the reported dimensions of the tumor) identified on baseline imaging were also collected. To establish potential delays in treatment between eras, we determined date of symptom onset through consultation notes and recorded date of first imaging, date of surgery and pathologic diagnosis, date of first oncologic consultation (medical or radiation oncology), and date of radiation initiation. During the COVID era, we additionally reviewed the initial surgical consult notes/operative notes, and radiation oncology/medical oncology consult and treatment notes for any mention of COVID as a consideration for treatment rationale or patient care. Treatment-related data included extent of surgery, dose of radiation prescribed, and prescription/completion of concurrent and/or adjuvant temozolomide. Survival data were recorded for all patients with overall survival defined as time from diagnosis to death of any cause. Living patients were censored based on last clinic follow-up date.

## 2.4. Statistical Analysis

Data analyses were performed using IBM SPSS version 27 (Markham, ON, Canada). Descriptive statistics were used to present the demographical information through counts, percentages, central tendency measures (mean and median), ranges, and variations. One-way ANOVA and chi-squared tests were used to compare baseline patient and treatment characteristics between eras. Kaplan–Meier (KM) survival estimates were used to assess progression-free survival (PFS) and overall survival (OS) and comparisons were made using the log-rank test. We used a Cox multivariate model to establish relationships between certain factors and survival. A conventional $p$-value $< 0.05$ was used to reject the null hypothesis.

## 3. Results

### 3.1. Patient Characteristics

One hundred and eighty-three patients with grade 4 glioma treated with radiotherapy at BC Cancer sites across the province were identified: ninety-one in the pre-COVID era and ninety-two in the COVID era. One hundred and sixty-four patients were retained for analysis: eighty-five patients in the pre-COVID era (93%) and seventy-nine patients in the COVID era (86%). In total, nineteen patients were excluded due to pathology other than grade 4 IDH wild-type glioblastoma or grade 4 IDH mutant astrocytoma (three in the pre-COVID era and eight in the COVID era), no pathologic diagnosis (three in the COVID era), loss to follow up (two in pre-COVID era and two in COVID era), and decline of treatment (one in pre-COVID era due to pregnancy).

Baseline characteristics are presented in Table 1. There were no statistically significant differences in patient characteristics between eras, including sex, number of comorbidities, and ECOG performance status. There was, however, a trend toward lower age of patients in the COVID era (mean age of 56 vs. 59 in the pre-COVID era, $p = 0.058$). When we further specifically examined the number of elderly patients treated, the percentage of patients aged 65 or older was 29.4% in the pre-COVID era compared to 16.5% in the COVID era, and this was a statistically significant difference ($p = 0.037$). Tumor characteristics including IDH mutation status and MGMT methylation status were not significantly different between eras. Of note however, there was a trend toward a greater average mass tumor diameter at first imaging in the COVID era compared to the pre-COVID era (4.6 cm vs. 4.1 cm, respectively), but this was not statistically significant ($p = 0.078$).

**Table 1.** Baseline characteristics of grade 4 glioma patients treated in the pre-COVID and COVID era. *ECOG* Eastern Cooperative Oncology Group; *IDH* isocitrate dehydrogenase; *MGMT* O$^6$-methylguanine DNA methyltransferase.

| | Pre-COVID Era | | COVID Era | | *p*-Value |
|---|---|---|---|---|---|
| | N | % | N | % | |
| Characteristic | | | | | |
| Number of patients | 85 | 100 | 79 | 100 | |
| Mean age (range) | 59 | (27–84) | 56 | (24–84) | 0.058 |
| Age $\geq$ 65 | 25 | 29.4 | 13 | 16.5 | 0.037 |
| Sex (F) | 25 | 29.4 | 28 | 35.4 | 0.255 |
| Comorbidities | | | | | 0.785 |
| 0 | 31 | 36.5 | 33 | 41.8 | |
| 1 | 21 | 24.7 | 18 | 22.8 | |
| $\geq$2 | 33 | 38.8 | 28 | 35.4 | |
| ECOG | | | | | 0.119 |
| 0 | 21 | 24.7 | 11 | 13.9 | |
| 1 | 36 | 42.4 | 42 | 53.2 | |
| $\geq$2 | 16 | 18.8 | 21 | 26.6 | |
| Unknown | 12 | 14.1 | 5 | 6.3 | |
| IDH mutation | | | | | 0.304 |
| Yes | 2 | 2.4 | 4 | 5.1 | |
| No | 78 | 91.8 | 70 | 88.6 | |
| Unknown | 5 | 5.9 | 5 | 6.3 | |
| MGMT methylation | | | | | 0.34 |
| Yes | 25 | 29.4 | 25 | 31.6 | |
| No | 27 | 31.8 | 34 | 43.0 | |
| Unknown | 33 | 38.8 | 20 | 25.3 | |
| Average mass diameter in cm | 4.1 | | 4.6 | | 0.078 |

*3.2. Treatment Details*

Table 2 shows the time to evaluation and treatment between eras. There was no statistically significant difference in the time from symptom onset (as estimated in initial consultation notes) to presentation (31 days in the pre-COVID era compared to 28 days in the COVID era, *p* = 0.751). The mean time from first imaging to surgery was the same between eras (13 days in both eras, *p* = 0.936). Time from surgery to first oncology consultation (medical or radiation oncology) was not significantly different (3 weeks in both eras, *p* = 0.768), and importantly, neither was time from surgery to initiation of radiation (6 weeks in both eras, *p* = 0.475).

**Table 2.** Treatment timelines for patients treated in the pre-COVID and COVID eras.

|  | Pre-COVID Era | COVID Era | *p*-Value |
|---|---|---|---|
|  | Mean | Mean |  |
| Time from symptom onset to imaging (days) | 31 | 28 | 0.751 |
| Time from imaging to surgery (days) | 13 | 13 | 0.936 |
| Time from surgery to onc consultation (weeks) | 3 | 3 | 0.768 |
| Time from surgery to RT (weeks) | 6 | 6 | 0.475 |

The extent of surgery between the two eras was significantly different (*p* = 0.037). Specifically, more patients were surgically managed with biopsy in the COVID era compared to the pre-COVID era (21.5% vs. 11%, respectively), and fewer patients were managed with gross total resection (56.5% vs. 36.7%, respectively). With respect to radiation treatment, more patients were treated with palliative course radiation of 25 Gy in five fractions in the COVID era (15.2%) compared to the pre-COVID era (9.4%). While only two patients were treated with hypofractionated RT (40–45 Gy in 15 fractions) in the pre-COVID era, no patients were treated with that regimen in the COVID era. However, there was no statistically significant difference in radiotherapy prescription between eras (*p* = 0.321). Furthermore, there was no significant difference in the use of concurrent chemotherapy (83.5% in the pre-COVID era vs. 79.7% in the COVID era, *p* = 0.268), prescription of adjuvant temozolomide (*p* = 0.374), or rate of completion of adjuvant temozolomide (43.9% in the pre-COVID era vs. 53.4% in the COVD era, *p* = 0.304) between eras (Table 3).

**Table 3.** Treatment characteristics of patients treated in the pre-COVID and COVID eras. *TMZ* temozolomide; *RT* radiotherapy.

|  | Pre-COVID Era | | COVID Era | | |
|---|---|---|---|---|---|
|  | N | % | N | % | *p*-Value |
| Extent of surgery |  |  |  |  | 0.037 |
| Gross total resection | 48 | 56.5 | 29 | 36.7 |  |
| Partial resection | 26 | 30.6 | 33 | 41.8 |  |
| Biopsy | 11 | 12.9 | 17 | 21.5 |  |
| Radiation treatment |  |  |  |  | 0.321 |
| 60 Gy | 74 | 87.1 | 65 | 82.3 |  |
| 40–45 Gy | 2 | 2.4 | 0 | 0.0 |  |
| 25 Gy | 8 | 9.4 | 12 | 15.2 |  |
| Unfinished | 1 | 1.2 | 2 | 2.5 |  |
| Concurrent TMZ with RT |  |  |  |  | 0.268 |
| No | 9 | 10.6 | 14 | 17.7 |  |
| Yes | 71 | 83.5 | 63 | 79.7 |  |
| Unfinished | 5 | 5.9 | 2 | 2.5 |  |
| Adjuvant TMZ prescribed |  |  |  |  | 0.374 |
| No | 28 | 32.9 | 21 | 26.6 |  |
| Yes | 57 | 67.1 | 58 | 73.4 |  |
| Adjuvant TMZ completed (when prescribed) |  |  |  |  | 0.304 |
| No | 32 | 56.1 | 27 | 46.6 |  |
| Yes | 25 | 43.9 | 31 | 53.4 |  |

*3.3. Direct Influence of COVID on Management during the COVID Era*

During the COVID era, we further evaluated whether COVID directly impacted treatment decision-making and patient care. In total, COVID was only noted to potentially impact patient care in five cases (6.3% of patients). One patient was worked up for COVID and underwent isolation prior to glioma diagnosis, potentially resulting in a delay in diagnosis. COVID was not mentioned as a factor in decision-making for surgical extent in any of the neurosurgical consult notes or operating notes. During adjuvant treatment, two patients were diagnosed with COVID during initial evaluation and treatment: one patient was diagnosed with COVID prior to starting radiation and was treated with a hypofractionated schedule due to resulting poor performance status, and the second patient had adjuvant temozolomide delayed due to COVID diagnosis. A fourth patient had a change in a radiation treatment plan from 40 Gy in 15 fractions to a palliative course of 25 Gy in 5 fractions due to the requirement for hospitalization during radiation and concerns with COVID exposure during hospitalization. The fifth patient had adjuvant temozolomide delayed by one week due to the COVID vaccine.

*3.4. Survival between Eras*

Progression-free survival (PFS) and overall survival (OS) were evaluated between each era, with a median follow-up of 12 months (range 1–46 months) for patients treated in the pre-COVID era, and 15 months (range 1–25 months) for patients treated in the COVID era. Median PFS was 7 months in the pre-COVID era (CI$_{95}$ 5.50–8.50 months) and 8 months in the COVID era (CI$_{95}$ 6.76–9.24 months) in the COVID era ($p = 0.347$). Median OS was 13 months in the pre-COVID era (CI$_{95}$ 10.3–15.7 months), and 16 months in the COVID era (CI$_{95}$ 11.6–20.4 months), and the difference was not statistically significant ($p = 0.189$, Figure 1). Cox multivariate analysis revealed a non-significant association between mass diameter and survival (HR 1.01, CI$_{95}$ 0.92–1.11). However, gross total resection significantly decreased the risk of death (HR 0.46, CI$_{95}$ 0.28–0.75), while the age of 65 or greater significantly increased the risk of death (HR 1.83, CI$_{95}$ 1.20–2.78).

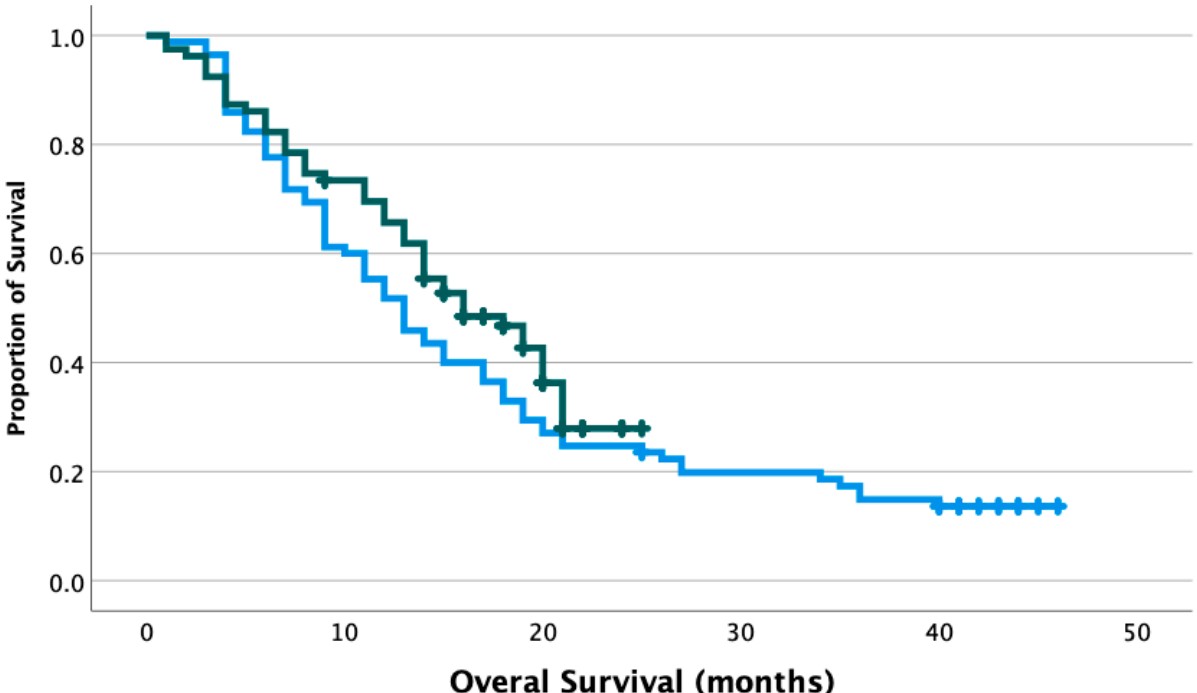

**Figure 1.** Overall survival of grade 4 glioma patients treated in the pre-COVID and COVID era. Blue line: patients treated in the pre-COVID era, median OS 13 months. Green line: patients treated in the COVID era, median OS 16 months. $p = 0.189$. Crosshatches represent censored outcomes.

## 4. Discussion

The COVID-19 pandemic caused significant strain on healthcare resources globally and influenced the care of all patients, including those with cancer [13–15]. In the field of neuro-oncology, numerous guidelines recommended the prioritization of high-grade gliomas in the face of potential severe strains on neurosurgical resources and recommended appropriate adjuvant therapy while trying to minimize the risk of COVID for both patients and healthcare providers [7,16]. To our knowledge, our study is the first to investigate the management and outcome of patients with grade 4 glioma during the COVID-19 pandemic in Canada.

We evaluated patients seen at BC Cancer sites across the province who received radiation as part of their adjuvant treatment. The pre-COVID and COVID-era cohorts had similar baseline tumor and patient characteristics, including the number of comorbidities and baseline performance status. We did, however, note a trend towards larger tumor diameter during the COVID era. Similar trends were observed for patients with other cancers during the COVID pandemic. One study reported a substantial increase in breast cancer patients presenting at an advanced stage compared to a pre-pandemic cohort in Brazil [17], while another showed increased morbidity from larger tumor size in patients with head and neck cancers [15]. These trends are thought to represent a delay in presentation to hospitals during COVID, perhaps due to fear of contracting COVID or general recommendations to avoid hospital visits when possible [18].

However, the mildly larger tumor diameter during the COVID era noted in our study was not statistically significant. Furthermore, we did not see a delay in time from symptom onset to first imaging between eras, which also aligns with the similarity in performance status at presentation between eras. This may be due to the unique presentation of high-grade brain tumors compared to other cancers, including acute neurologic deficits prompting more urgency. This is also in keeping with a prior single-institution study from New York that showed that though the number of neurosurgical consultations was significantly lower during the first month of COVID relative to a pre-pandemic time period, the percentage of patients with brain tumors within these consults stayed stable, suggesting that these patients were not more or less likely to present for care [19].

Of note, we also did not identify any delays in management during the COVID era compared to the pre-COVID era, and only one patient had a possible delay in diagnosis due to suspicion of having COVID. Time from initial imaging to surgery, surgery to oncology consultation, and time from surgery to initiation of radiation treatment were all statistically similar between eras. This is in contrast to several studies evaluating the effect of the pandemic on time to diagnosis and management of more common cancers that may not present as acutely. For example, one systematic review of forty-three studies of colorectal cancers treated in the first year of the pandemic showed that the number of diagnostic procedures and treatment of colorectal cancers had decreased, with associated delays in all treatments including surgery, radiation, and chemotherapy [20]. Another study based in the US identified a significant decrease and delays in the screening of several cancers (including breast, colon, prostate, and lung), reductions in new patient and established patient evaluations, and decreased billing frequency of physician-administered oncology services (ex. chemotherapy) [21].

With respect to surgical management, we observed significantly fewer gross total resections and more biopsies performed during the COVID era compared to the pre-COVID era. Conversely, a retrospective study by Amoo et al. [22] performed at a tertiary care center in Ireland showed no reduction in the proportion of radical resections for patients with malignant brain tumors during the pandemic. In the practice recommendation by Bernhardt et al., they highlighted that maximal safe surgery should still be considered a priority for high-grade gliomas. They did, however, recognize that within a crisis phase of the pandemic, the surgical intent may be altered in efforts to minimize lengthy hospital stays during a period of limited resources and higher risk of COVID [7]. In our population, though it is difficult to ascertain the rationale for surgical extent with the limitations of

our retrospective analysis, a review of neurosurgical consult and operative notes did not suggest that the COVID pandemic played a role in clinical decision-making regarding the extent of resection. This suggests that biopsy was likely the most appropriate management option for other reasons. Similarly, in a single-center retrospective analysis from France, while routine neurosurgical care was maintained during COVID without compromising patient access to the required treatment, the authors additionally noted that glioblastoma cases were more aggressive than the previous year and more biopsies were performed because they were the recommended management choice [23].

Fortunately, COVID seemed to only impact adjuvant treatment for a small minority of patients, and of 79 patients treated in the COVID era, only 2 were diagnosed with COVID-19 infection during evaluation and treatment. There were no statistically significant differences in radiation, concurrent chemoradiotherapy, or adjuvant chemotherapy prescription or completion between eras. Interestingly, though hypofractionated regimens of radiation were recommended for glioma patients during COVID to minimize the risk of exposure [10], there were no patients treated with the hypofractionated schedule of 40 Gy in 15 fractions during the COVID era. Instead, more patients were treated with a palliative course of 25 Gy in five fractions (though this was not statistically significant), despite a similar performance status between eras and a lower average age of patients during the COVID era. This also accounts for the slight reduction in concurrent chemotherapy used in the COVID era, as concurrent temozolomide is rarely prescribed alongside palliative intent radiation [24]. With respect to adjuvant temozolomide, the rate of completion of the prescribed course was similar between eras. Early in the pandemic, treatment with temozolomide was thought to potentially increase the risk of contracting severe COVID-19 infections due to possible myelotoxicity and lymphopenia [7], which is associated with increased mortality from COVID [25]. No patients were diagnosed with COVID during adjuvant treatment, suggesting that temozolomide did not increase the risk of contracting the infection.

For patients with other cancers, the long-term impact of delays and alterations in cancer care will not be fully appreciated for years to come, but several studies highly suggest that delays in diagnosis, resulting in a more advanced stage at the time of treatment, and delays to treatment will likely result in worsened overall survival [17,20,21,26]. However, for grade 4 gliomas, which have a relatively short prognosis, our follow-up period is sufficient to evaluate median overall survival and accurately analyze how differences between the pre-COVID and COVID eras influenced survival. Given the reduction in gross total resections during the COVID era, with all other treatments being equal, we would have expected a reduction in OS during the COVID era based on historical data showing better outcomes with gross total resection compared to partial resection or biopsy [27]. This was confirmed with Cox multivariate analysis showing decreased risk of death with gross total resection in our study. This, however, is not reflected in the OS observed, as there is no statistically significant difference in survival between eras. In fact, there is a slight trend towards improved survival in the COVID era, with a median OS of 16 months vs. 13 months in the pre-COVID era.

What likely accounts for the similar, and slightly improved survival despite fewer gross total resections in the COVID era is the statistically significant reduction in patients over the age of 65 treated during the COVID era compared to the pre-COVID era. This is again confirmed by Cox multivariate analysis showing an increased risk of death with an age over 65. Of note, our analysis only includes patients who received radiation as part of their adjuvant treatment, so it is unclear if fewer elderly patients were being offered the same care as they would have pre-pandemic (whether it be opting not to pursue surgery or biopsy, or not being offered radiation), or if they were not presenting to hospital in general. Protective isolation was recommended globally during the early stage of the pandemic, especially for the elderly. Consequently, a study based in Ireland showed a 16% reduction in the presentation of patients over the age of 70 in a single emergency department, with a 20% reduction in patients presenting with serious stroke or cardiac complaints [28]. Similarly, in Italy, there was a 25% decrease in general emergency department visits by elderly

patients in 2020 compared to the year prior, but the proportion of patients presenting by ambulance with more severe conditions requiring hospitalization has increased sharply since March 2020 [1].

Furthermore, for older adults with cancer specifically, the International Society of Geriatric Oncology recommended that therapeutic decisions should favor the most effective approach with the least risk of adverse outcomes, which may include omitting treatments such as surgery, radiation therapy, or chemotherapy when benefits are considered marginal [29]. For elderly patients with grade 4 glioma, though the standard of care involves short-course hypofractionated radiation with concurrent and adjuvant temozolomide [30], there is significant variability in the management of this heterogenous population, and prognosis remains poor. Post-operative adjuvant treatment can range from the standard of care to radiation alone (whether it be hypofractionated radiation or palliative course radiation), to temozolomide alone, to best supportive care, and is largely dependent on performance status and $O^6$-methylguanine DNA-methyltransferase (MGMT) methylation status [31]. Therefore, if the elderly patients in our cohort were presenting later with more severe symptoms, they may have been less likely to be offered treatment.

Notable limitations to our study include the small sample size, the retrospective nature of the study limiting the ability to determine causality, and the fact that we only included patients who received radiation, therefore excluding patients who were diagnosed with grade 4 gliomas and were not offered adjuvant treatment or were offered only chemotherapy alone. However, our analysis did include patients treated at all BC Cancer sites and therefore provides a comprehensive analysis of patients from different resource settings in one of the most populated provinces in the country.

## 5. Conclusions

Thus far, there have only been a few reported real-world analyses looking at the provision of care for neuro-oncology patients during the COVID-19 pandemic, and outcomes for these patients during the pandemic have varied based on the patient population and geographical location [19,22,32,33]. These studies, however, did not focus on grade 4 gliomas, which account for the most prevalent and aggressive primary brain tumor affecting adults. Our study found that the COVID-19 pandemic, fortunately, did not delay access to care or impact the survival of patients with grade 4 gliomas in British Columbia, despite a reduction in gross total resections performed. Our data are somewhat confounded however by the significant reduction in elderly patients treated during the pandemic, highlighting the need to optimize access to care for all patients with high-grade gliomas. It is critical that patients, regardless of their age, are encouraged to seek care when they have new neurologic concerns. Given that COVID-19 continues to be prevalent, and the emergence of new aggressive variants remains a possibility, these results provide reassurance that appropriate and timely care can be provided for glioma patients during periods of limited hospital resources.

**Author Contributions:** Conceptualization, M.C. and B.T.; data curation, M.C. and G.A.; formal analysis, M.C.; methodology, M.C., A.N. and B.T.; supervision, R.H., A.N. and B.T.; writing—original draft, M.C. and G.A.; writing—review and editing, R.H., A.N. and B.T. All authors have read and agreed to the published version of the manuscript.

**Funding:** This research received no external funding.

**Institutional Review Board Statement:** The study was conducted in accordance with the Declaration of Helsinki and approved by the Institutional Review Board of BC Cancer/Systemic Therapy-Vancouver (BC Cancer) (protocol H21-02144, 18 August 2021).

**Informed Consent Statement:** Patient consent was waived due to the retrospective nature of the study.

**Data Availability Statement:** The data presented in this study are available on request from the corresponding author. The data are not publicly available due to privacy and ethical restrictions.

**Conflicts of Interest:** The authors declare no conflict of interest.

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
