# Peer review of "Treatment Patterns and Outcomes of Patients with Grade 4 Glioma Treated with Radiation during the COVID-19 Pandemic"

_curroncol, doi:10.3390/curroncol30030234_

Round 1
Reviewer 1 Report
Thank you for the opportunity to review "Treatment patterns and outcomes of patients with grade 4 glioma treated with radiation during the COVID-19 pandemic". This paper highlights the important issue of brain tumor management during the COVID-19 pandemic.
Is likely that the results are affected both by the reduced rate of gross total resections in the pre-COVID era and by patients excluded from the study in the COVID era because they did not have access to surgical or radiotherapy treatment. Surely the study would be even more interesting if it were possible to retrieve data from all patients diagnosed with HHG (including those with presumptive radiological diagnosis who did not undergo biopsy). However, if this were not possible, the limitations of the study are already clearly expressed by the authors.
Since it is known from the literature that extent of resection correlates with survival, it would be useful to better illustrate how the resection classes (GTR, STR, and biopsy) were defined and on what basis the type of surgical approach was chosen.
Author Response
Review 1 comments:
“Surely the study would be even more interesting if it were possible to retrieve data from all patients diagnosed with HHG (including those with presumptive radiological diagnosis who did not undergo biopsy). However, if this were not possible, the limitations of the study are already clearly expressed by the authors.
Response: Unfortunately, we are not able to access this information within the limits of our database. I appreciate that the reviewer recognizes that we addressed this in our limitations.
“Since it is known from the literature that extent of resection correlates with survival, it would be useful to better illustrate how the resection classes (GTR, STR, and biopsy) were defined and on what basis the type of surgical approach was chosen.”
Response:
I have addressed how the resection classes were defined in the methods (lines 87-90). With respect to what basis the surgical approach was chosen, it is difficult to determine the rationale for extent of resection from the limits of a retrospective review.
However, we did refer back to the operative notes and initial surgical consults during the COVID era to determine if COVID was mentioned as part of the rationale for extent of surgery (lines 117-120 in methods). I have also included a new section in results to highlight the role that COVID may have played in treatment decision making- for surgery and adjuvant therapy (lines 202-225).
Reviewer 2 Report
Authors present a retrospective study on radiation therapy treatment of grade 4 gliomas in British Columbia during COVID-19 pandemic, with comparison of 85 patients in period 2020-2021 with 79 patients in period 2018-2019; beside the fact that there were less older patients and more patients who underwent biopsy in the covid era, there were no other differences.
Low number of patients and retrospective character of the study are its drawbacks. The number of patients is low to make conclusions, but the study is not without merit as it shows a single center experience. It is not clear that there is any connection between the pandemic and diminished surgical invasiveness (more biopsies than resection in COVID-19 era). One very important aspect has been left out - complication rate (of surgery and of radiaton therapy) as well as possible delay (in days) of time from diagnosis to surgery and from surgery to radiation therapy in both cohorts (important due to possible delay due to pandemic in the covid-era). Did any of these patients actually had also COVID? I suggest to include a group of cerebral metastases who underwent radiation therapy treatment, with the same method (pre-covid and covid-cohort) to investigate if there are similar trends in this group too.
Author Response
Reviewer 2 comments:
“One very important aspect has been left out - complication rate (of surgery and of radiaton therapy) as well as possible delay (in days) of time from diagnosis to surgery and from surgery to radiation therapy in both cohorts (important due to possible delay due to pandemic in the covid-era).”
Response:
Due to the limitations in our database, we are not able to address the complication rate.
However, we did address possible delays which were already presented in Table 2. To make these results more apparent, I have added more detail to the results section under section 3.2 Treatment Details (lines 167-175), and expanded the discussion (lines 279-292)
“Did any of these patients actually had also COVID?”
Response:
We have addressed this in the new results section 3.3 Direct influence of COVID in management during the COVID era (lines 202-225).
“I suggest to include a group of cerebral metastases who underwent radiation therapy treatment, with the same method (pre-covid and covid-cohort) to investigate if there are similar trends in this group too.”
Response:
Respectfully, this is beyond the scope of this article which aims to focus on a more homogeneous population of grade 4 gliomas. The goal is to determine treatment and outcome differences in this population specifically, as it represents the most common and most aggressive primary brain malignancy in adults.
Round 2
Reviewer 2 Report
Authors have sufficiently responded to reviewer remarks.